# Analysis of Bacterial Phosphorylcholine-Related Genes Reveals an Association between Type-Specific Biosynthesis Pathways and Biomolecules Targeted for Phosphorylcholine Modification

Yuan Zhang,[a] Freda E.-C. Jen,[a] Jennifer L. Edwards,[b,c] Michael P. Jennings[a]

[a]Institute for Glycomics, Griffith University, Southport, Australia
[b]Center for Microbial Pathogenesis, The Abigail Wexner Research Institute at Nationwide Children's Hospital, Columbus, Ohio, USA
[c]Department of Pediatrics, The Ohio State University, Columbus, Ohio, USA

Yuan Zhang and Freda E.-C. Jen are joint first authors and contributed equally to this work. Author order was determined based on their contributions.

**ABSTRACT** Many bacterial surface proteins and carbohydrates are modified with phosphorylcholine (ChoP), which contributes to host mimicry and can also promote colonization and survival in the host. However, the ChoP biosynthetic pathways that are used in bacterial species that express ChoP have not been systematically studied. For example, the well-studied Lic-1 pathway is absent in some ChoP-expressing bacteria, such as *Neisseria meningitidis* and *Neisseria gonorrhoeae*. This raises a question as to the origin of the ChoP used for macromolecule biosynthesis in these species. In the current study, we used *in silico* analyses to identify the potential pathways involved in ChoP biosynthesis in genomes of the 26 bacterial species reported to express a ChoP-modified biomolecule. We used the four known ChoP biosynthetic pathways and a ChoP transferase as search terms to probe for their presence in these genomes. We found that the Lic-1 pathway is primarily associated with organisms producing ChoP-modified carbohydrates, such as lipooligosaccharide. Pilin phosphorylcholine transferase A (PptA) homologs were detected in all bacteria that express ChoP-modified proteins. Additionally, ChoP biosynthesis pathways, such as phospholipid *N*-methyltransferase (PmtA), phosphatidylcholine synthase (Pcs), or the acylation-dependent phosphatidylcholine biosynthesis pathway, which generate phosphatidylcholine, were also identified in species that produce ChoP-modified proteins. Thus, a major finding of this study is the association of a particular ChoP biosynthetic pathway with a cognate, target ChoP-modified surface factor; i.e., protein versus carbohydrate. This survey failed to identify a known biosynthetic pathway for some species that express ChoP, indicating that a novel ChoP biosynthetic pathway(s) may remain to be identified.

**IMPORTANCE** The modification of bacterial surface virulence factors with phosphorylcholine (ChoP) plays an important role in bacterial virulence and pathogenesis. However, the ChoP biosynthetic pathways in bacteria have not been fully understood. In this study, we used *in silico* analysis to identify potential ChoP biosynthetic pathways in bacteria that express ChoP-modified biomolecules and found the association between a specific ChoP biosynthesis pathway and the cognate target ChoP-modified surface factor.

**KEYWORDS** phosphorylcholine, ChoP, phosphatidylcholine, phosphoethanolamine, bacterial virulence

Address correspondence to Michael P. Jennings, m.jennings@griffith.edu.au.

The authors declare no conflict of interest.

Phosphorylcholine (ChoP) is a small zwitterionic molecule and can be found in Gram-positive bacteria (e.g., *Streptococcus pneumoniae*, the pneumococcus [1]) and Gram-negative bacteria (e.g., *Haemophilus influenzae* (2) and commensal *Neisseria* spp. [3]). ChoP modifications predominantly occur on glycoconjugates or proteins located on the cell surface

**TABLE 1** Bacteria expressing ChoP modification

| ChoP-modified structure | Organism(s) | Colonization site | Reference(s) |
|---|---|---|---|
| ChoP-modified glycan(s)[a] | | | |
| Teichoic acid | *Streptococcus pneumoniae* R36A | Respiratory tract | 5 |
| | *Streptococcus oralis* Uo5 | Respiratory tract | 34 |
| | *Streptococcus mitis* NCTC10712 | Respiratory tract | 77 |
| Capsular polysaccharide | *Streptococcus pneumoniae* type 15 | Respiratory tract | 78 |
| | *Streptococcus pneumoniae* type 32F | Respiratory tract | 79 |
| | *Erysipelothrix rhusiopathiae* | Skin | 38 |
| Lipopolysaccharides | *Haemophilus influenzae* Rd | Respiratory tract | 80 |
| | *Haemophilus haemolyticus* | Respiratory tract | 30 |
| | Commensal *Neisseria* | Respiratory tract | 3 |
| | *Avibacterium paragallinarum* | Respiratory tract | 37 |
| | *Pasteurella multocida* AP161 | Nasopharynx or gastrointestinal tract | 81 |
| | *Histophilus somni* 738 | Respiratory tract | 7 |
| | *Proteus mirabilis* O18 | Respiratory, intestinal, or urinary tract | 42 |
| | *Morganella morganii* O1 | Intestinal tract | 43 |
| | *Fusobacterium nucleatum* strain 25586 | Respiratory or intestinal tract | 82 |
| Phosphoglycolipid | *Mycoplasma fermentans* | Respiratory or urinary tract | 83 |
| ChoP-modified protein[a] | | | |
| Pilus | *Neisseria meningitidis* | Respiratory tract | 13, 21 |
| | *Neisseria gonorrhoeae* | Ocular, nasopharyngeal, or anal mucosa | 46, 84 |
| Fimbrial protein Flp 1 | *Aggregatibacter actinomycetemcomitans* | Respiratory tract | 9 |
| Porin D | *Acinetobacter baumannii* | Skin or respiratory tract | 10 |
| ChoP mimic[a] | | | |
| Elongation factor Tu | *Pseudomonas aeruginosa* | Respiratory tract | 29 |
| Unknown ChoP-modified structure[b] | | | |
| | *Bacillus* spp. | Gastrointestinal tract | 85 |
| | *Gemella haemolysans* | Respiratory tract | 85 |
| | *Micrococcus* spp. | Skin | 85 |
| | *Actinomyces viscosus* | Oropharynx | 86 |
| | *A. gerencseriae* | Oropharynx | |
| | *Lactococcus* spp. | Respiratory tract | 85 |
| | *Corynebacterium jeikeium* | Skin | 85 |
| | *Streptococcus pyogenes* | Pharynx, anus, or genital mucosa | 87 |

[a]Structural evidence for ChoP modification is described in the cited reference.
[b]Studies where MAb recognition of ChoP by TEPEC15 is the only evidence for ChoP expression.

of bacteria. To date, there are 14 bacterial species in which ChoP glycoconjugates, such as ChoP-modified wall teichoic acid (WTA), lipoteichoic acid (LTA) (4–6), and lipooligosaccharides/lipopolysaccharides (LOS/LPS) (2, 3, 7), have been characterized (Table 1). Examples of ChoP-modified proteins include the type IV fimbriae (pili) of *Neisseria meningitidis* and *Neisseria gonorrhoeae* (8), Flp fimbriae of *Aggregatibacter actinomycetemcomitans* (9), and porin D of *Acinetobacter baumannii* (10). It has been demonstrated that the surface expression of ChoP promotes pneumococcal (11) and nontypeable *H. influenzae* (NTHi) (12) adhesion to human airway cells. ChoP expressed on these pathogenic bacteria plays a crucial role in mediating bacterial adherence and invasion of airway epithelial cells via the platelet-activating factor receptor (PAFr) (12, 13). Furthermore, ChoP facilitates *H. influenzae* colonization of the human nasopharynx (12) and maturation of biofilms (14). The benefits of ChoP modification in bacterial pathogenesis and its impact on the modulation of host immunity have been thoroughly reviewed by Clark and Weiser (15).

Four distinct pathways are described for the biosynthesis of ChoP that is destined for incorporation into a surface biomolecule (Fig. 1). The LOS/LPS core (Lic-1) pathway is a well-known ChoP biosynthetic pathway (Fig. 1a); the Lic-1 pathway biosynthetic enzymes are encoded by the *licABCD* operon (2, 16). Apart from environmental free choline, choline-containing molecules derived from the host cell lipid metabolism serve as the potential sources for the Lic-1 pathway (17–19). In the absence of free choline, *H. influenzae* utilizes glycerophosphodiester phosphodiesterase (GlpQ) to acquire choline from the respiratory tract epithelial cells (18, 19). The choline permease LicB is required for choline absorption

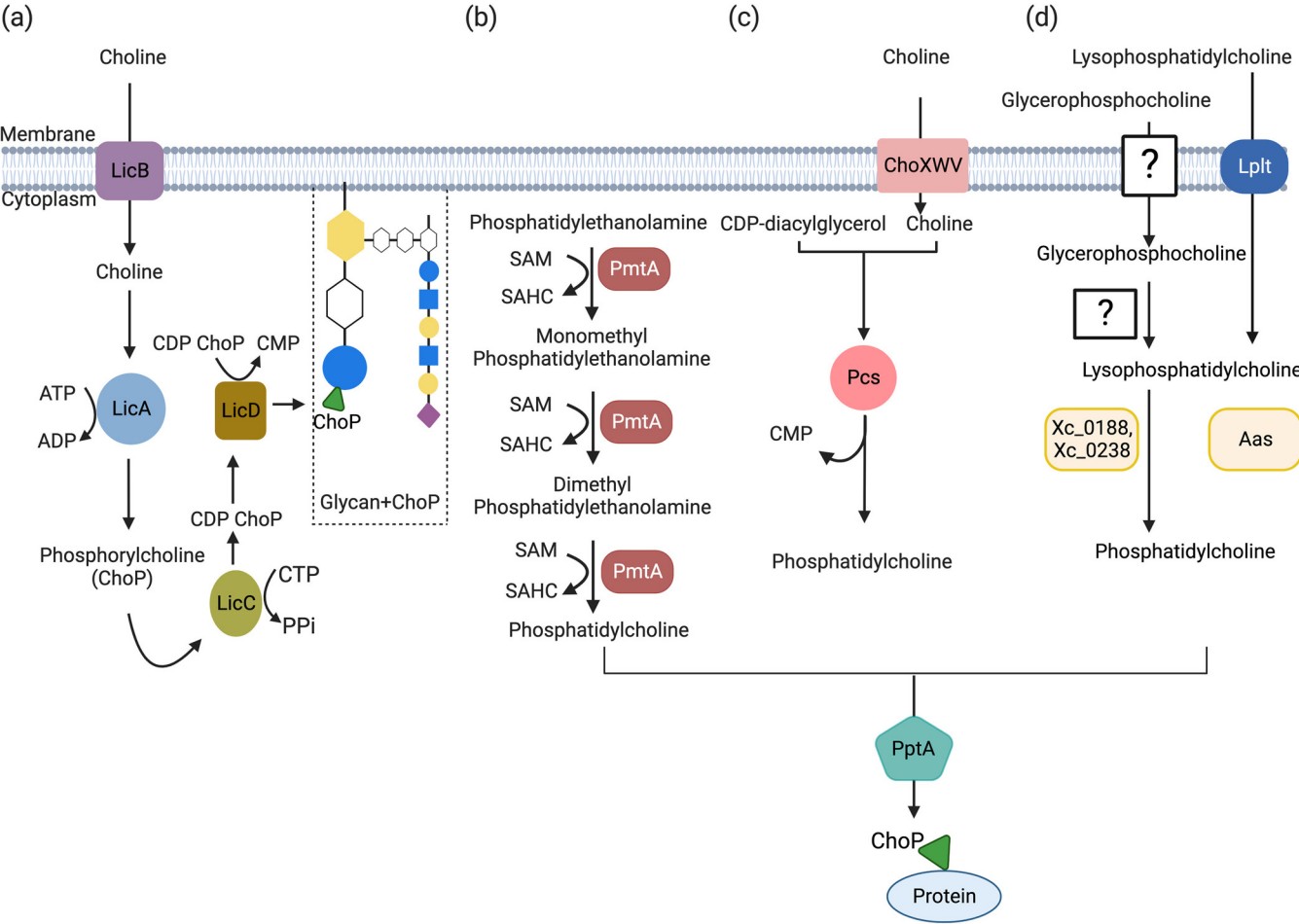

**FIG 1** Characterized pathways for biosynthesis of candidate ChoP donor molecules for macromolecule modification by ChoP. (a) Lic-1 pathway. LicB takes up choline from the environment, which is then converted to CDP-choline by LicA and LicC. LicD transfers the activated CDP-ChoP to a glycan structure, such as LOS or teichoic acid. (b) PmtA pathway. Phosphatidylethanolamine is methylated at three positions by PmtA to form phosphatidylcholine. (c) Pcs pathway. Pcs enzymes catalyze the condensation of choline with CDP-diacylglycerol to phosphatidylcholine (PC), releasing a CMP molecule. (d) Acylation-dependent PC biosynthesis pathway. Abbreviations: SAHC, S-adenosylhomocysteine; SAM, S-adenosylmethionine; PPi, pyrophosphate; Lplt, lysophospholipid transporter; Aas, acyltransferase-acyl carrier protein synthase.

and transport in bacteria (2, 17, 20). Choline is phosphorylated by the choline kinase LicA in the cytoplasm to create ChoP. LicC is a phosphorylcholine cytidylyltransferase that uses CTP and ChoP to convert ChoP into CDP-choline. A choline phosphotransferase, called LicD, then transfers ChoP to the glycans structures, such as WTA, LTA, and LOS/LPS (2).

Unlike the *S. pneumoniae*, *H. influenzae*, or commensal *Neisseria* examples described above, in *N. meningitidis* and *N. gonorrhoeae*, the addition of ChoP to pilin is not generated by the enzymes encoded by the *lic* genes (3, 21). To date, pilin phosphorylcholine transferase A (PptA) is the only enzyme that has been identified as being responsible for mediating any step in ChoP modification on pilin of *N. meningitidis* and *N. gonorrhoeae* (22, 23); that is, no ChoP biosynthetic pathway has been described. A homopolymeric tract of guanosine residues exists in the coding region of *pptA*, and the alterations in the length of this tract correlate with the phase-variable expression of ChoP on pilin (13, 23). The Lic-1 ChoP biosynthetic pathway generates a CDP-ChoP intermediate that acts as a donor molecule for the LicD ChoP transferase, analogous to the nucleotide sugar donors typically used by glycosyltransferases. In contrast, other ChoP biosynthetic pathways generate a ChoP lipid, phosphatidylcholine (PC), as a donor molecule for ChoP transfer. Currently, there are three well-studied biosynthetic pathways for the production of PC in bacteria, including phospholipid *N*-methyltransferase (PmtA [Fig. 1b]), phosphatidylcholine synthase (Pcs [Fig. 1c]), and acylation-dependent (Fig. 1d) PC biosynthesis pathways. In the PmtA pathway, PmtA produces PC by the sequential addition of a methyl group from

*S*-adenosylmethionine (SAM) to phosphatidylethanolamine (24), starting with monomethyl phosphatidylethanolamine (MMPE), dimethyl phosphatidylethanolamine (DMPE), and phosphatidylcholine (Fig. 1b). Whereas the Lic-1 pathway is dependent on exogenous choline, the PmtA pathway is independent of exogenous choline, as it synthesizes choline *de novo*.

Like the Lic-1 pathway, exogenous choline is required in the Pcs pathway. Choline transporters such as ChoXWV take up the choline from the environment (25), and Pcs combines CDP-diacylglycerol (CDP-DAG) and choline to produce PC and the by-product CMP. Similarly, the recently described acylation-dependent PC biosynthesis pathway uses exogenous choline in the form of glycerophosphocholine or lysophosphatidylcholine. This pathway has been described for only three bacterial species (26, 27), but it is well-characterized in yeast. For a more complete description of the Pcs, PmtA, and acylation-dependent PC biosynthesis pathways, see the recent review by Zhang et al. (28).

This study aimed to identify the ChoP biosynthetic pathway that is used by each of the 26 species of bacteria that have been reported to express ChoP. This was achieved by bioinformatic analysis of ChoP synthesis-related genes (as presented above [Fig. 1]) present in genomes of bacteria expressing ChoP on surface biomolecules. We also included the analysis of an enzyme called EftM (29), which synthesizes a ChoP structural mimic in *Pseudomonas aeruginosa*. This posttranslational modification of a lysine to add additional methyl groups on elongation factor protein, EF-Tu, results in a structural mimic of ChoP that reacts with the ChoP-specific monoclonal antibody TEPC15. Collectively, this new information will enable a correlation between the class of ChoP biosynthetic pathway and the target biomolecules that are modified by ChoP and will facilitate a more complete understanding of the factors controlling the expression of this important virulence factor.

## RESULTS AND DISCUSSION

**Survey for the presence of Lic-1 pathway enzymes in ChoP-expressing bacteria reveals an association between Lic-1 and expression of ChoP-modified glycoconjugates.** We first used the protein sequence of Lic-1 pathway enzymes, including LicA, LicB, LicC, and LicD (Fig. 1), in *H. influenzae* as the query in a Basic Local Alignment Search Tool for protein sequences (BLASTp) search against the NCBI nonredundant (nr) protein database. Figure 2 shows a schematic representation of the proteins and genes involved in the Lic-1 pathway in *H. influenzae* and the summary of Lic-1 pathways protein found in other bacteria. The presence of Lic-1 pathway enzymes corresponds well with the bacterial species that are reported to express carbohydrate-linked ChoP, including representatives of *Haemophilus haemolyticus* (30), *S. pneumoniae* (5), and commensal *Neisseria* (3) (Fig. 2). The homologs of Lic-1 enzymes that were identified have ~30% identity to the LicABCD search sequences and are listed in Fig. 2. Moreover, the LicA, -B, -C, and -D homologs in *H. haemolyticus* (31), *Histophilus somni* (32), *S. pneumoniae* (16, 33), *Streptococcus oralis* (34), *Streptococcus mitis* (35), *Pasteurella multocida* (36), commensal *Neisseria* species (3), *Avibacterium paragallinarum* (37), *Erysipelothrix rhusiopathiae* (38), and *Mycoplasma fermentans* (39–41) have previously been shown to mediate the synthesis of glycoconjugates, such as WTA-, LTA-, and LOS-ChoP structures. In this study, we have identified for the first time the presence of the Lic-1 locus in *Proteus mirabilis* (42) and *Morganella morganii* (43), both of which express ChoP-modified glycan structures. In summary, all bacteria containing carbohydrate-linked ChoP had Lic-1 pathway enzyme homologs.

Interestingly, the Lic-1 protein family appears to be restricted to organisms with carbohydrate-linked ChoP, and it has not been found in *N. meningitidis*, *N. gonorrhoeae* (8), *A. actinomycetemcomitans* (9), or *A. baumannii* (10). For each of these 4 species, ChoP-modified protein structures are reported. *Gemella haemolysans* (44), *Actinomyces gerencseriae*, *Actinomyces viscosus*, and *Bacillus cereus* all contain Lic-1 pathway enzymes and, therefore, may have typical WTA/LTA/LOS ChoP structures, even though ChoP-modified glycoconjugates have not been reported for these organisms to-date.

**Identification of PptA in bacteria containing ChoP modification.** We next focused our analysis on the organisms known to express ChoP but that have no Lic-1 locus. This list included four bacteria reported to express ChoP-modified proteins (Fig. 3) and four organisms with no defined ChoP-modified structure (Fig. 4). Pathogenic *Neisseria* species express

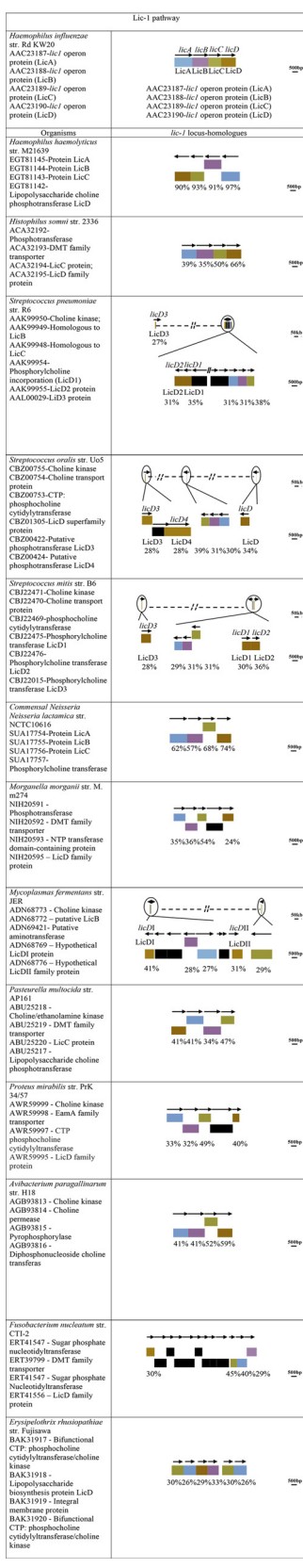

**FIG 2** Enzyme homologs identified in bacteria containing ChoP-carbohydrates. The arrangements of the genes and proteins in the Lic-1 pathway are indicated, with LicA, LicB, LicC, and LicD depicted in blue, purple, green, and brown, respectively. Organisms and strain name are shown to the left. Homologs that were found are indicated next to the stain name with accession numbers. The arrow represents the gene orientation. The percent sequence identity with Lic-1 pathway proteins in *H. influenzae* is displayed below each gene box.

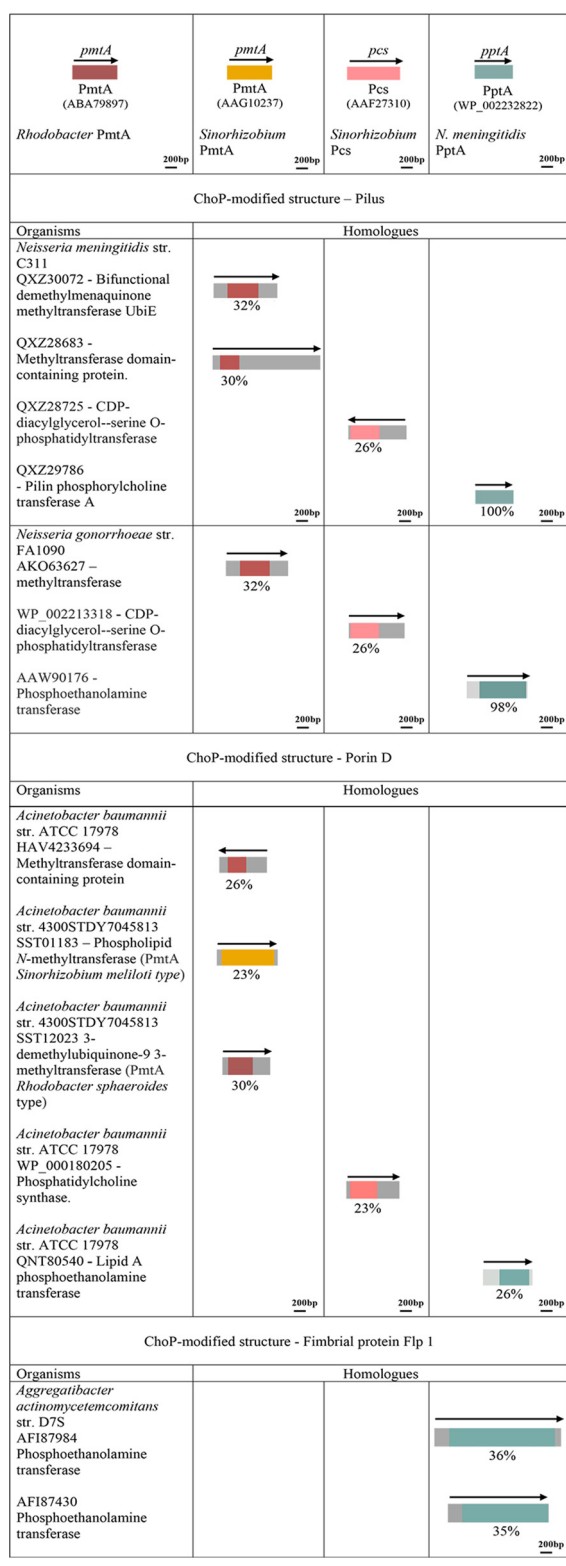

**FIG 3** Enzyme homologs identified in bacteria containing ChoP-modified protein. The known PmtA of *R. sphaeroides* (61), PmtA of *S. meliloti* (62), Pcs of *S. meliloti* (62), and characterized PptA (13, 23) with accession numbers are depicted in red, yellow, pink, and green, respectively. Organisms and strain names are shown to the left. Homologs that were found are indicated next to the stain name with accession numbers. The arrow represents the gene orientation. The percent sequence identity with *R. sphaeroides* PmtA, *S. meliloti* PmtA, *S. meliloti* Pcs, and *N. meningitidis* PptA is displayed below each gene box.

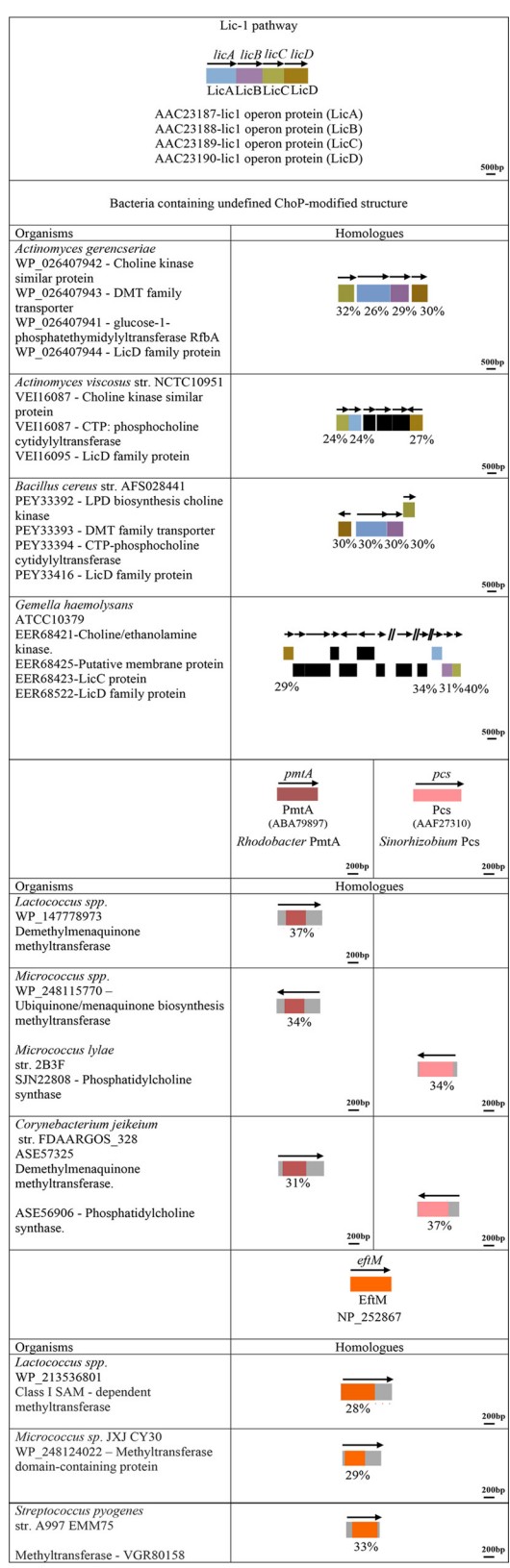

**FIG 4** Enzyme homologs identified in bacteria containing undefined ChoP-modified structures. The arrangements of genes and proteins in the Lic-1 pathway are illustrated, with LicA, LicB, LicC, and LicD depicted in blue, purple, green, and brown, respectively. The known PmtA of *R. sphaeroides* (61), PmtA of *S. meliloti* (62), Pcs of *S. meliloti* (62), characterized *N. meningitidis* PptA (13, 23), and *P. aeruginosa* EftM (29), with their respective accession numbers, are shown in red, yellow, pink, green, and orange. Organisms

ChoP-modified pilin. In *N. meningitidis*, PptA recognizes the amino acid sequence [153]CRDASDAS[160] present within the C terminus of the pilin subunit protein, PilE (45), and modifies Ser[157] and Ser[160] with ChoP (13). In *N. gonorrhoeae*, PptA is responsible for the ChoP or phosphoethanolamine (PE) modification at Ser[68] and Ser[156] (22, 46). To examine if the PptA transferase is present in other organisms containing ChoP-decorated proteins, or bacteria expressing unknown ChoP structures, we used the C311 PptA (NCBI:protein accession no. QXZ29786) protein sequence from the fully annotated sequenced *N. meningitidis* C311 genome (47) as a query. In each genome, protein similarity searches (BLASTp) were used to identify PptA homologs.

We observed that all organisms with a ChoP-modified protein, including *A. actinomycetemcomitans* and *A. baumannii*, have PptA homologs (Fig. 3). Previously, it was discovered that Flp fimbriae of *A. actinomycetemcomitans* D7S are modified with ChoP and that two PptA homologs can be detected in this bacterium (9). Consistent with this observation, we discovered two open reading frames (ORFs) in *A. actinomycetemcomitans* with high sequence identities with PptA (36%). The two PptA-like ORFs are not solely in *A. actinomycetemcomitans* D7S (ORF1_AFI87984 and ORF2_AFI87430); 90% of strains have two PptA-like proteins, such as ORF1_TYB21792 and ORF2_TYB20790 from strain HK_907. In *A. baumannii*, six PptA-like ORFs were detected (Fig. 5), with some strains, such as *A. baumannii* 4300STDY7045813, having two or three PptA homologs. Ab_ORF1, Ab_ORF2, and Ab_ORF3 all have high sequence identities (30% to 34%) with PptA, whereas the identities among Ab_ORF4, Ab_ORF5, and Ab_ORF6 were low (26%).

It should be noted, nonetheless, that PptA belongs to the alkaline phosphatase superfamily and is closely related to a number of PE transferases responsible for modifying LPS with PE. PptA shares homology with two *N. meningitidis* PE transferases, LptA and Lpt3 (48), and also shows structural homology to *Escherichia coli* EptB/EptA and *A. baumannii* PmrC (22). Moreover, *N. gonorrhoeae* PptA is reported to contribute to the decoration of pilin with both ChoP and PE (46). In light of these observations, we cannot rule out the possibility that the role of PptA homologs discovered in this study may also include PE modification.

To analyze the evolutionary relationship between these PptA-like ORFs and the known PptA and PE transferases, and to gain insight into the functionality of PptA-like ORFs in *A. actinomycetemcomitans* and *A. baumannii*, a phylogenetic tree was constructed using the neighbor-joining method with MEGA X (Fig. 5). This analysis revealed two clades, the segregation of which depends on the substrate specificity of the PE transferase. One of the clades is represented by EptA (49), PmrC (50, 51), LptA (52), Lpt3 (53, 54), and Lpt6 (48), known to catalyze the addition of PE to the lipid A of LPS. The second clade is represented by EptB (55), which modifies the 3-deoxy-D-manno-octulosonic acid (Kdo) of LPS with PE. PptA clustered with EptB but did not appear to be closely related to it, implying that they have distinct functional properties. However, the ORF2 of *A. actinomycetemcomitans* and *A. baumannii* shared a node with EptB, and Aggr.a_ORF2 clustered together with EptB on the same branch. This indicates that these PptA-like ORFs may not act as ChoP transferases but rather may transfer PE. In contrast, PptA ORF1 of *A. actinomycetemcomitans* (Aggr.a_ORF1) clustered with *N. meningitidis and N. gonorrhoeae* PptA on the same node, suggesting that PptA functions for Aggr.a_ORF1 are probable. As shown in Fig. 5, in *A. baumannii*, only ORF1 and ORF2 exhibit relatively close proximity to PptA, while the remaining ORFs are grouped with other PE transferases, represented by EptA. Additionally, it is noteworthy that *A. baumannii* ATCC 17978, which contains ChoP-modified porin D (10), only displays the clustering of PptA-like ORF5 with PmrC, among the identified ORFs. PptA-like ORF5 (NCBI:protein accession no. QNT80540) is annotated as a PE transferase, and deletion of this gene could result in the loss of lipid A modification (50). Thus, these findings raise doubts regarding whether the

**FIG 4** Legend (Continued)
and strain names are indicated to the left, and homologs that were identified are noted next to the stain name, along with accession numbers. Gene orientation is represented by arrows, and the percent sequence identity is shown below each gene box.

## Tree scale: 0.2 ⊢━━┤

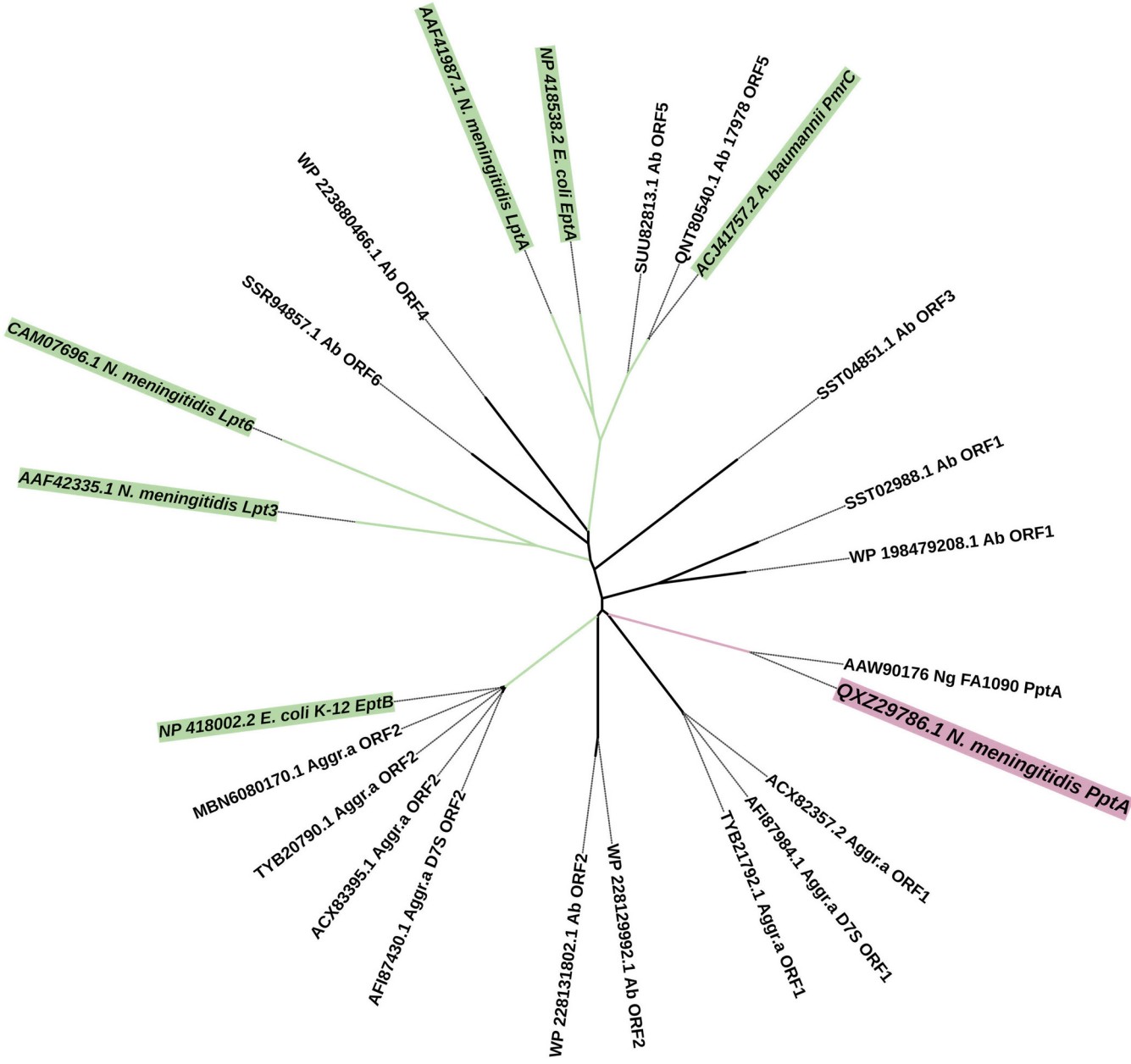

**FIG 5** Unrooted phylogenetic tree of PptA and PptA-like ORFs. The protein sequences of PptA-like ORFs were used to construct a neighbor-joining tree with characterized PptA (13, 23), EptA (49), EptB (55), PmrC (50, 51), LptA (52), Lpt3 (53, 54), and Lpt6 (48). *N. meningitidis* PptA is depicted in pink. *E. coli* EptB, *E. coli* EptA, *N. meningitidis* Lpt3, *N. meningitidis* Lpt6, *N. meningitidis* LptA, and *A. baumannii* PmrC are depicted in green. Distances between sequences are expressed as 0.2 change per amino acid residue. Ng, *N. gonorrhoeae*; Ab, *A. baumannii*; Aggr.a, *A. actinomycetemcomitans*. Accession numbers of the sequences of the proteins are shown.

PptA-like ORF of *A. baumannii* functions in a manner similar to that of PptA in *N. meningitidis*, which is known to act as a ChoP transferase. However, there is a possibility that PptA homologs found in *A. baumannii* ATCC 17978 could show bifunctional activities by participating in both ChoP and PE modification processes. Alternatively, PptA homolog-mediated PE modification may be followed by ChoP synthesis via PE methylation. ChoP is PE with three methyl groups. In parasites such as *Plasmodium falciparum* (56) and *Caenorhabditis elegans* (57), PE methyltransferase transfers three methyl groups to the amine of PE, resulting in the production of ChoP.

**Identification of PC biosynthesis pathways in bacteria containing ChoP protein modification.** There remains a gap in our knowledge as to what the ChoP donor used by PptA might be and the pathway(s) required for ChoP donor biosynthesis prior to pilin posttranslational modification. In this regard, phosphatidylcholine (PC) could be a potential ChoP donor. The headgroups of phospholipids, such as phosphatidylglycerol and phosphatidylethanolamine, are used for the biosynthesis of phosphoglycerol and phosphoethanolamine (58, 59). Cipollo et al. also found that *Caenorhabditis elegans* uses PC other than CDP-choline as a donor for the biosynthesis of a ChoP glycoprotein (60). PptA is similar in sequence and structure to phosphoethanolamine transferases like Lpt3, which uses phosphatidylethanolamine as a precursor to produce PE in *N. meningitidis* (53, 54). It is possible that PC may be a donor for ChoP-modified macromolecule biosynthesis. The corresponding well-known pathway in bacteria is the phospholipid *N*-methyltransferase (PmtA) and phosphatidylcholine synthase (Pcs). Recently, a further, acylation-dependent PC biosynthesis pathway was proposed as a novel PC biosynthesis pathway (Fig. 1).

**(i) PmtA and Pcs pathways.** The representative PmtA protein sequences from *Rhodobacter sphaeroides* (NCBI:protein accession no. ABA79897) (61) and *Sinorhizobium meliloti* (AAG10237) (62) were used as queries in a BLASTp search against *N. meningitidis*, *N. gonorrhoeae*, *A. actinomycetemcomitans*, *A. baumannii*, and three organisms (a *Micrococcus* sp., a *Lactococcus* sp., and *Corynebacterium jeikeium*) with undefined ChoP-modified structures. As shown in Fig. 3, we identified two *R. sphaeroides* PmtA homologs with 30% and 32% identities in the *N. meningitidis* C311 strain that have ChoP posttranslational modification on pilin (13). *Rhodobacter* PmtA ORFs from *N. gonorrhoeae* shared 32% identities with *R. sphaeroides* PmtA. Certain *A. baumannii* stains contain both *R. sphaeroides* and *S. meliloti* PmtA ORFs with high sequence identities (30% to 34%) (Fig. 6). However, the ATCC 17978 strain only possessed a *Rhodobacter* PmtA-like protein, which had low sequence identity to the queried sequences. PmtA homologs were not detected in *A. actinomycetemcomitans*. In contrast, PmtA homologs from the *Micrococcus* sp., *Lactococcus* sp., and *Corynebacterium jeikeium* all shared high sequence identities (31% to 37%) with *R. sphaeroides* PmtA (Fig. 4).

Previous studies have reported that in some bacteria, a single PmtA protein is not capable of catalyzing the 3-fold methylation of PE to generate PC (Fig. 1) and that some PmtA enzymes have distinct substrate specificities (63). *Bradyrhizobium japonicum* contains four Pmt proteins (PmtA, PmtX1, PmtX3, and PmtX4), and only the *R. sphaeroides* PmtA-like PmtA and PmtX1 can produce PC via subsequent methylation (64). PmtX3, PmtX4, and PmtA of *Xanthomonas campestris* (27) prefer to synthesize MMPE or DMPE. Therefore, we analyzed the evolutionary relationship between the putative PmtA ORFs and the known PmtA in bacteria (Fig. 6). Notably, phylogenetic analysis revealed that only *S. meliloti* ORF1, *R. sphaeroides* ORF1, and ORF2 from *A. baumannii* clustered with *R. sphaeroides* PmtA or *S. meliloti* PmtA. However, whether *A. baumannii* stains that contain homologs of *S. meliloti* ORF1 and/or *R. sphaeroides* ORF1 and ORF2, such as *A. baumannii* AB32_M and 4300STDY7045804, have a ChoP-modified structure(s) has not been confirmed. The *N. meningitidis* homolog of *R. sphaeroides* ORF1 (NMB1270) has been studied previously (23), but ChoP expression is unaltered with inactivation of this protein. Consequently, it remains challenging to determine the precise roles of these PmtA homologs in *A. baumannii* and *N. meningitidis*. It is possible that these PmtA homologs do not play a role in ChoP biosynthesis and/or this PmtA-like ORF may act like PmtX3, PmtX4, or PmtA of *X. campestris* and are responsible for MMPE or DMPE production.

Different from the PmtA pathway, the Pcs pathway requires choline uptake from the environment to synthesize phosphatidylcholine. A BLASTp search showed that Pcs homologs could be found in *N. meningitidis*, *N. gonorrhoeae*, and *A. baumannii*. These Pcs homologs shared low sequence identities (23% to 36%) with *S. meliloti* Pcs (Fig. 3). Since Pcs shows high identity with phosphatidylglycerol phosphate synthase (PgsA) and phosphatidylserine synthase (PssA) (65), some of the Pcs homologs found in other bacteria may not exhibit phosphatidylcholine synthase activity. As shown in Fig. 7, Pcs homologs from *N. meningitidis*, *N. gonorrhoeae*, and *A. baumannii* ATCC 17978 were clustered with *E. coli* PgsA on the same branch, and they presumably have the same function.

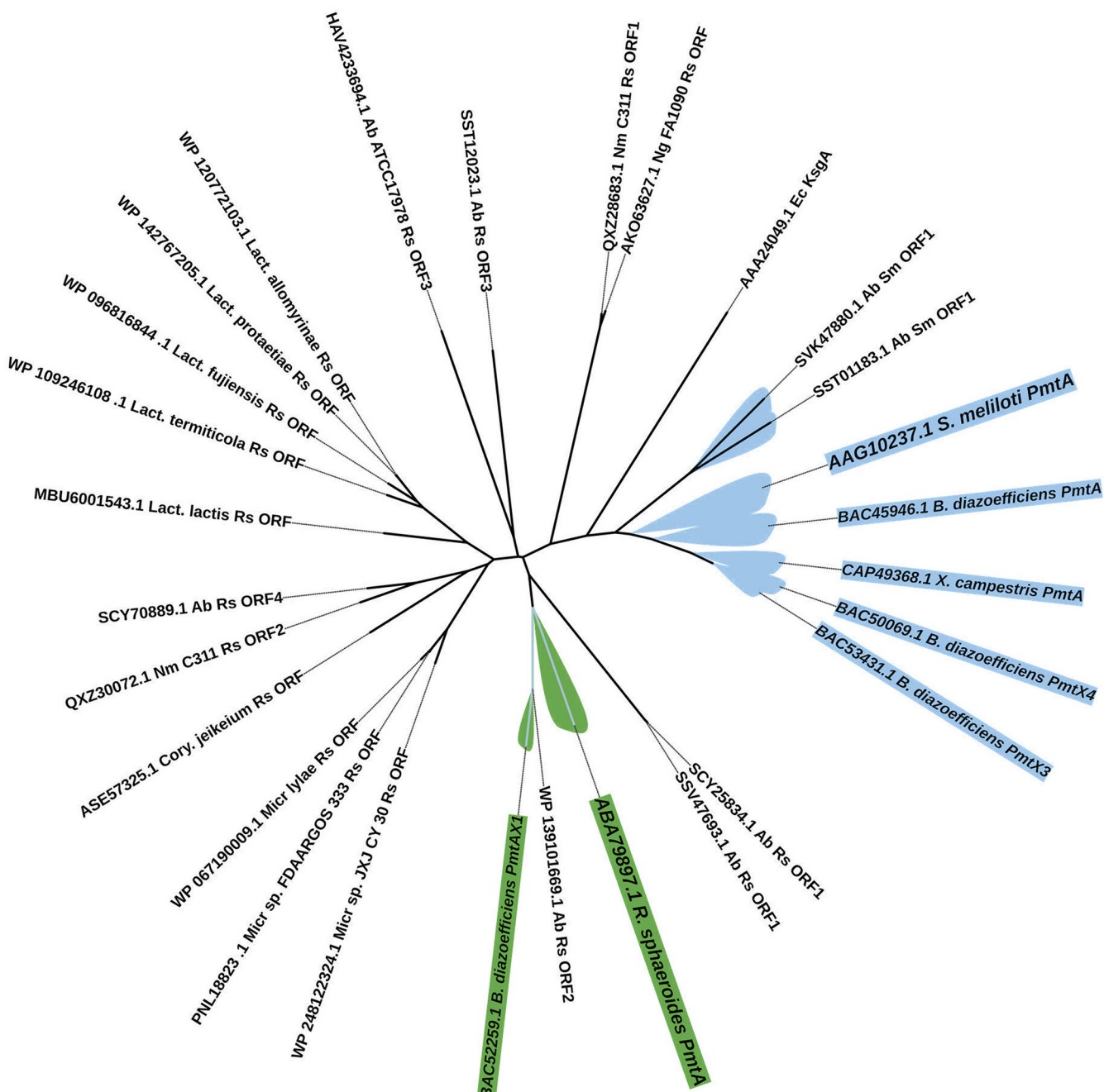

**FIG 6** Unrooted phylogenetic tree of PmtA and PmtA-like ORFs. The protein sequences of PmtA-like ORFs were used to construct a neighbor-joining tree with known PmtA of *R. sphaeroides* (61), *S. meliloti* (62), PmtA, PmtX1, PmtX3, and PmtX4 of *B. japonicum* (64), and PmtA of *X. campestris* (27). *R. sphaeroides* PmtA and *B. japonicum* PmtAX1 are shown in green. *S. meliloti* PmtA, *B. japonicum* PmtA, PmtAX4, and PmtAX3, and *X. campestris* PmtA are highlighted in blue. *R. sphaeroides* PmtA-like ORFs are abbreviated as Rs ORFs. *S. meliloti* PmtA-like ORFs are referred to as Sm ORFs. Distances between sequences are expressed as 0.2 change per amino acid residue. Rs, *R. sphaeroides*; Sm, *S. meliloti*; Nm, *N. meningitidis*; Micr, *Micrococcus* spp.; Lact, *Lactococcus* spp.; Cory, *C. jeikeium*. Accession numbers of the sequences of the proteins are shown.

**(ii) Acylation-dependent PC biosynthesis pathway.** The acylation-dependent PC biosynthesis pathway has been reported only for the Gram-negative bacterium *E. coli* (26), the plant pathogen *X. campestris* (27), and the Gram-positive pathogens *S. pneumoniae*, *S. mitis*, and *S. oralis* (66). *X. campestris* converts exogenous glycerophosphocholine (GPC) to lysophosphatidylcholine (lyso-PC) via two acylation reactions on GPC (Fig. 1). To date, two *X. campestris* acyltransferases (Xc_0188 and Xc_0238) that perform

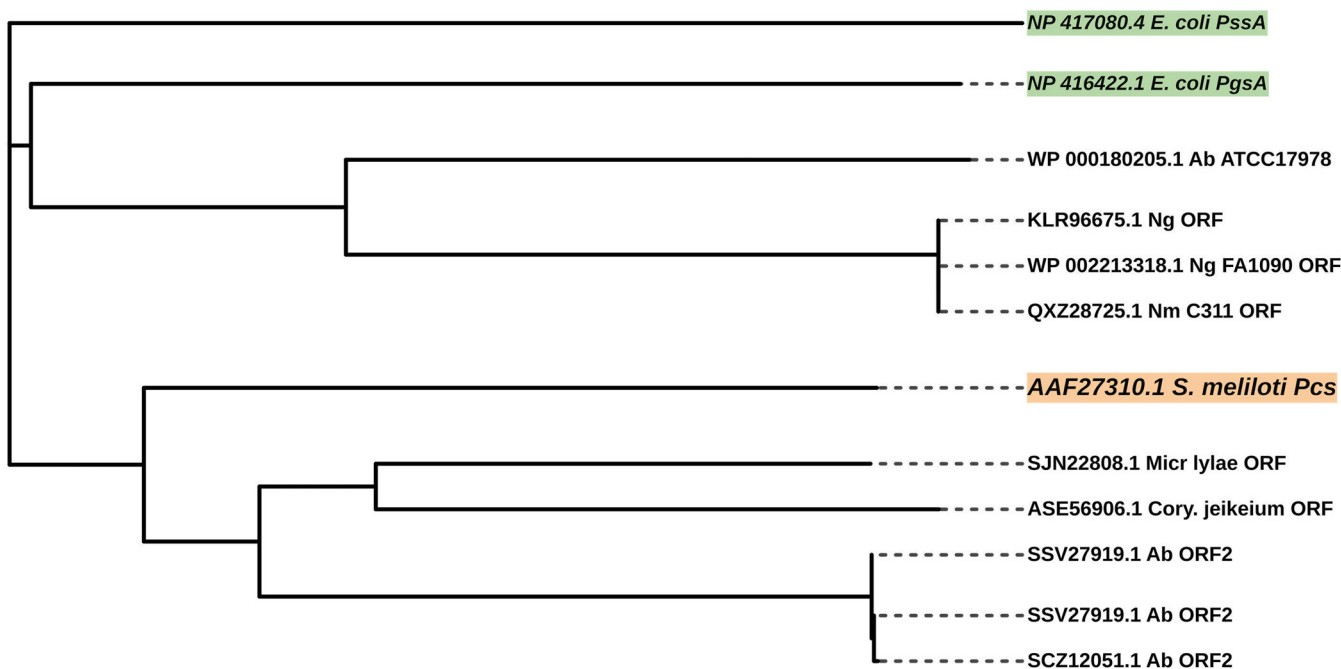

**FIG 7** Unrooted phylogenetic tree of Pcs and Pcs-like ORFs. The protein sequences of Pcs-like ORFs were used to construct a neighbor-joining tree with known Pcs of *S. meliloti* (62) (orange boxes) and PssA and PgsA of *E. coli* (65) (green boxes). Distances between sequences are expressed as 0.1 change per amino acid residue. Accession numbers of the sequences of the proteins are shown.

the second acylation from lyso-PC to PC have been identified (27). Additionally, *E. coli* can produce PC with the action of the lysophospholipid transporter Lplt and the acyltransferase-acyl carrier protein synthase Aas. Lplt has the capacity to take up lyso-PC, and then Aas converts lyso-PC to PC (26). We used Xc_0188 (NCBI:protein accession no. AAY47275) (27), Xc_0238 (NCBI:protein accession no. AAY47325) (27), Aas (WP_000899054) (26), and Lplt (WP_000004616) (26) as the query in a BLASTp search against *N. meningitidis*, *N. gonorrhoeae*, *A. baumannii*, *A. actinomycetemcomitans*, *Micrococcus* spp., *Lactococcus* spp., *C. jeikeium*, *Streptococcus pyogenes*, *S. pneumoniae*, *S. mitis*, and *S. oralis*. For evolutionary analysis and grouping of enzymes involved in the acylation-dependent PC biosynthesis pathways, a multiple-sequence alignment using the ClustalW algorithm was created using MEGA X software. The phylogenetic trees based on these alignments were constructed using the neighbor-joining method (Fig. 8 and Fig. 9).

Regarding the Lplt/Aas system, we identified Aas ORFs in *S. pneumoniae* and *A. actinomycetemcomitans*, both of which clustered with Aas of *E. coli* on the same main clade with a bootstrap value of 100 (Fig. 8). The bootstrap value indicates the percentage of the replicate trees that recovered that specific clade. Lplt ORFs were also found in these two organisms and shared a very robust node with Lplt (Fig. 9) with a high bootstrap value of 100. Interestingly, in *A. actinomycetemcomitans*, Aas ORF and Lplt ORF belong to one protein; the *S. pneumoniae aas* and *lplt* genes seem to be included in the same operon. This observation is consistent with the *lplt* and *aas* gene arrangement in *E. coli* (67). In addition, it is reported that *S. pneumoniae* can use lyso-PC to synthesize PC (66), suggesting that *S. pneumoniae* and *A. actinomycetemcomitans* might use the Lplt/Aas system to produce PC.

Similar to *X. campestris*, *S. pneumoniae* and *S. mitis* can use GPC for PC biosynthesis (66). As shown in Fig. 8, the Xc_0188-like ORFs of *N. meningitidis*, *N. gonorrhoeae*, *S. pneumoniae*, and *A. baumannii* clustered with Xc_0188 on the same main branch and formed the Xc_0188 group. Xc_0238 ORFs of *N. meningitidis*, *N. gonorrhoeae*, and *A. actinomycetemcomitans* clustered together and shared a node with the Xc_2038 clade that clustered with the *S. mitis* ORF and *A. baumannii* ORFs. Acyltransferases are poorly defined in bacteria. Therefore, it is reasonable to infer that *N. meningitidis*, *N. gonorrhoeae*, and *A. baumannii*

Tree scale: 0.1 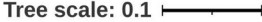

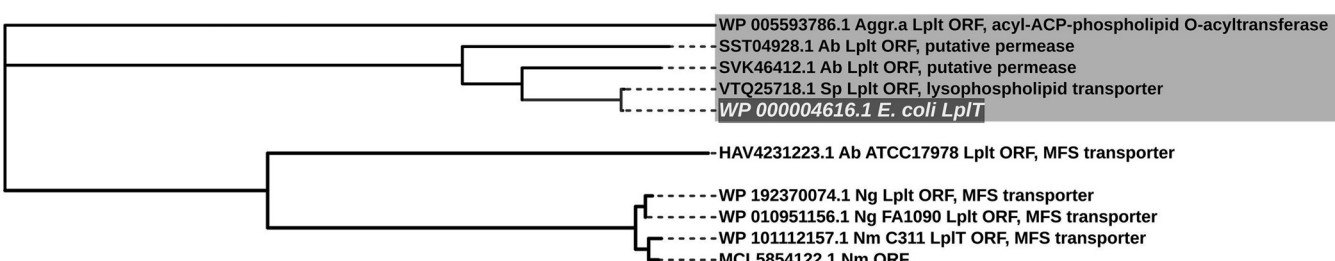

**FIG 8** Phylogenetic analysis of a number of known and predicted acyltransferases involved in PC biosynthesis. The protein sequences of Aas, Xc_0188, and Xc_0238-like ORFs were used to construct a neighbor-joining tree with known Aas of *E. coli* (26), Xc_0188 of *X. campestris* (27), and Xc_0238 of *X. campestris* (27). They are depicted in dark purple, dark red, and dark yellow, respectively. The tree shows three different lyso-PC acyltransferase families, represented by shaded clusters with the following colors: purple for the Aas family, pink for the Xc_0188 family, and yellow for the Xc_0238 family. Sp, *S. pneumoniae*; St, *S. mitis*; So, *S. oralis*. Distances between sequences are expressed as 0.1 change per amino acid residue. Accession numbers of the sequences of the proteins are shown.

may have the capacity to synthesize PC via an acylation-dependent pathway, in which Xc_0188 and Xc_0238 ORFs are responsible for acylation of lyso-PC to PC. However, there is still a knowledge gap related to the transporter for GPC uptake and the GPC-specific acyltransferase(s).

**ChoP mimic biosynthesis pathway.** A ChoP modification is reported for the EF-Tu elongation factor protein of *Pseudomonas aeruginosa* (68). However, mass spectrometry analysis of purified, native EF-Tu revealed that this appears to contain a ChoP mimic with

Tree scale: 0.1

**FIG 9** Phylogenetic analysis of a number of known and predicted Lplt transporters involved in PC biosynthesis. The protein sequences of Lplt-like ORFs were used to construct a neighbor-joining tree with known Lplt of *E. coli* (26). The shaded cluster represents the Lplt family. Distances between sequences are expressed as 0.1 change per amino acid residue. Accession numbers of the sequences of the proteins are shown.

a different mass than ChoP (29). In this regard, a novel methyltransferase (EftM) methylates the lysine three times to form a novel trimethyl structure (29). This trimethyl-modification on lysine has a chemical structure similar to that of the ChoP epitope and is recognized by anti-ChoP antibodies (29). In the present study, EftM homologs were identified in the genome of *Lactococcus* spp., *Micrococcus* spp., and *S. pyogenes*, which shared 28% to 33% identity (Fig. 4). Whether these EftM homologs exhibit methyltransferase activity, resulting in the generation of a protein-associated ChoP-like epitope in these bacteria, has not been examined.

**Conclusions.** ChoP modification of surface-exposed virulence factors is a key feature of many bacterial pathogens. Understanding the biosynthetic pathways for ChoP biosynthesis is key to understanding the role of this modification in virulence. ChoP modification in bacteria is not limited to carbohydrates in that several surface-exposed, proteinaceous, bacterial virulence factors are now known to be decorated with ChoP. In this study, our analysis of the well-characterized ChoP biosynthetic pathways revealed that the Lic-1 pathway seems to be a common pathway for ChoP-linked glycoconjugates, such as LOS/LPS, WTA, and LTA. In contrast, bacteria that produce ChoP-bearing proteins, such as *N. meningitidis*, *N. gonorrhoeae*, *A. baumannii*, and *A. actinomycetemcomitans*, appear to lack the *lic-1* operon but do contain PptA transferase homologs. There is currently no evidence to suggest the coexistence of the Lic-1 pathway and PptA enzymes within bacterial species. The nature of the pathways that provide the ChoP donor molecule for PptA-dependent transfer of ChoP to ChoP-modified proteins remains to be identified and is the subject of our ongoing studies.

## MATERIALS AND METHODS

**Selection of bacteria with ChoP modification and biosynthetic enzymes.** Twenty-six organisms that are reported to have ChoP modification were selected for this study (Table 1). All core enzymes involved in ChoP biosynthesis (LicABCD [16, 69], PmtA [70], Pcs [70], Xc_0188 [27], Xc_0238 [27], Aas [26], Lplt [26], and EftM [29] enzymes) or modification (ChoP transferases, PptA [13, 23], and LicD [16, 69]) were used to query the database.

**Homology analysis of enzymes in bacteria with a role in ChoP modification.** The accession numbers of the ChoP enzyme sequences used as search terms are listed in Fig. 2 and Fig. 6 to 9. A Basic Local Alignment Search Tool for protein sequences (BLASTp) search (71) using the selected ChoP enzymes as queries against the nonredundant (nr) protein sequence database (72) of different organisms was performed to detect homologs of ChoP enzymes in different pathways. All BLASTp searches were performed using the default parameters (73). Proteins were assumed to be homologous if a significant hit was found to be less than $10^{-5}$ (E value cutoff). Results were then filtered according to the identity threshold of 20%. The best-hit matches for each organism were retained for downstream analysis.

**Multiple-sequence alignments and neighbor-joining phylogenetic analysis.** Amino acid sequences retrieved by BLASTp search in the nr protein database were first aligned with MUSCLE (Codons) available in MEGA software (version X) (74) using the default algorithm, and the UPGMA (unweighted pair group method using average linkages) was selected as the cluster method. Phylogenetic tree construction and phylogenetic analysis were performed in MEGA X software (74) with the neighbor-joining method (75) and the Poisson model. The bootstrap test was used to validate the trees, with 1,000 bootstrap replicates, and the cutoff value was 50%. All trees were visualized and modified using iTOL v.5 (76).

## ACKNOWLEDGMENTS

This work was supported by Australian National Health and Medical Research Council (NHMRC) program grant 1071659 and Principal Research Fellowship 1138466 to M.P.J. and Ideas grant 2001210 to F.E.-C.J., a Griffith University International Postgraduate Research Scholarship (GUIPRS) to Y.Z., and National Institutes of Health, National Institute of Allergy and Infectious Diseases, grant R01AI134848 to J.L.E. and M.P.J.

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
