## [Reviewer comments · Microbiology Spectrum]

Microbiology Spectrum

Analysis of bacterial phosphorylcholine related genes reveals an association between the type-specific biosynthesis pathways and the biomolecules targeted for phosphorylcholine modification

Yuan Zhang, Freda Jen, Jennifer Edwards, and Michael Jennings

Corresponding Author(s): Michael Jennings, Griffith University

Review Timeline:

Submission Date:	April 14, 2023
Editorial Decision:	May 13, 2023
Revision Received:	June 4, 2023
Accepted:	June 8, 2023

Editor: Xiaoyu Tang

Reviewer(s): Disclosure of reviewer identity is with reference to reviewer comments included in decision letter(s). The following individuals involved in review of your submission have agreed to reveal their identity: Kaan Çeylan (Reviewer #1); Jeffrey N. Weiser (Reviewer #2)

Transaction Report:

DOI: <https://doi.org/10.1128/spectrum.01583-23>

May 13, 2023

Prof. Michael P. Jennings
Griffith University
Gold Coast Campus, QLD 4222
Australia

Re: Spectrum01583-23 (Analysis of bacterial phosphorylcholine related genes reveals an association between the type-specific biosynthesis pathways and the biomolecules targeted for phosphorylcholine modification)

Dear Prof. Michael P. Jennings:

Link Not Available

Sincerely,

Xiaoyu Tang

Journals Department
Reviewer comments:

Reviewer #1 (Comments for the Author):

The review of your work has been completed and I do not have any additional suggestions.
I wish good work

Reviewer #2 (Public repository details (Required)):

lots of large datasets appropriate for a depository

Reviewer #2 (Comments for the Author):

General: This submission provides an updated analysis of pathways involved in the expression of phosphorylcholine (ChoP) on the surface of bacteria. As this host-like structure appears on the cell surface in multiple ways and on many diverse species, this in silico analysis provides a useful roadmap for this genetic complexity. I have relatively few comments.

- 1) Abstract: Line 24-5. It would be clearer if you delete although.....is well studied, this.
- 2) Abstract: Line 29 bacterial species instead of bacteria
- 3) Abstract: Line 34. Not clear what Similarly refers to?
- 4) It would be helpful to point out those structures/species for which there is structural evidence for ChoP expression as opposed to mAb recognition, which as pointed out in the case of *Pseudomonas aeruginosa* is putative.
- 5) It might be useful to cite the review on bacterial ChoP by SE Clark (IAI 2013) which discusses some of the biological advantages of cell surface ChoP expression.
- 6) It might help the readers to explain the structural relationship between PE and ChoP.
- 7) Are there examples of species with both Lic1 and PptA homologs/pathways?
- 8) The potential sources of ChoP (i.e. choline uptake v. biosynthesis) could be explained much more clearly. This should include environmental free choline and the transfer of choline from host intermediates involved in phosphatidylcholine biosynthesis (see Fan X, Mol Micro 2001, 2003) which is not spelled out.
- 9) In Table 1, consider adding the primary site within the host where each species is typically found.
- 10) A brief discussion of the comparison to phosphocholine biosynthesis to generate PC in eukaryotes would be a useful addition.

Staff Comments:

Preparing Revision Guidelines

Please return the manuscript within 60 days; if you cannot complete the modification within this time period, please contact me. If you do not wish to modify the manuscript and prefer to submit it to another journal, please notify me of your decision immediately so that the manuscript may be formally withdrawn from consideration by Microbiology Spectrum.

Dear Prof Tang,

Thank you for your letter and for the reviewer's comments concerning our manuscript entitled "Analysis of bacterial phosphorylcholine related genes reveals an association between the type-specific biosynthesis pathways and the biomolecules targeted for phosphorylcholine modification". These comments have been very valuable in improving our revised manuscript. We have responded to each comment, detailed in the point by point response below and in the manuscript, as indicated.

Reviewer #1 (Comments for the Author):

The review of your work has been completed and I do not have any additional suggestions. I wish good work.

Response

We thank the reviewer.

Reviewer #2 (Public repository details (Required)):

Lots of large datasets appropriate for a depository.

Response

We disagree. The study comprises a series of blast searches using search terms defined by accession number of the sequence of the protein being used as a search term (noted in the main text and table), and key observations from the search results defined (with accession number and % identity) by lists of relevant proteins in the tables. There are no large datasets that are appropriate to deposit. Any person can use the approach we describe to replicate our data.

Reviewer #2 (Comments for the Author):

General: This submission provides an updated analysis of pathways involved in the expression of phosphorylcholine (ChoP) on the surface of bacteria. As this host-like structure appears on the cell surface in multiple ways and on many diverse species, this in silico

analysis provides a useful roadmap for this genetic complexity. I have relatively few comments.

1) Abstract: Line 24-5. It would be clearer if you delete although.....is well studied, this.

Response

Lines 26-28: As suggested, this sentence has been modified and now reads, “For example, the well-studied Lic-1 pathway is absent in some ChoP-expressing bacteria, such as *Neisseria meningitidis* and *Neisseria gonorrhoeae*.” (See pg. 1, line 26-28.)

2) Abstract: Line 29 bacterial species instead of bacteria

Response

Line 31: As suggested, the word “bacteria” has been modified to “bacteria species” and now reads “In the current study, we used in silico analyses to identify the potential pathways involved in ChoP biosynthesis in genomes of the twenty-six bacteria species reported to express a ChoP-modified biomolecule.” (See pg. 1, line 29-31.)

3) Abstract: Line 34. Not clear what Similarly refers to?

Response

Line 36: As per the suggestion, the word "Similarly" was inadequate and has been replaced with "Additionally". The context now reads as follows: “Pilin phosphorylcholine transferase A (PptA) homologs were detected in all bacteria that express ChoP-modified proteins. Additionally, ChoP biosynthesis pathways, such as phospholipid N-methyltransferase (PmtA), phosphatidylcholine synthase (Pcs), or the acylation-dependent phosphatidylcholine biosynthesis pathway, which generate phosphatidylcholine, were also identified in species that produce ChoP modified proteins.” (See pg. 2, line 34-39.)

4) It would be helpful to point out those structures/species for which there is structural evidence for ChoP expression as opposed to mAb recognition, which as pointed out in the case of *Pseudomonas aeruginosa* is putative.

Response

Line 637: The Table 1 of this manuscript has headings which indicates the source of the information about ChoP expression. We have added footnotes to clarify structural studies vs antibody only studies as follows, “Footnotes: ¹ Structural evidence for ChoP modification is described in the cited reference, ² studies where mAb recognition of ChoP by TEPEC15 is the only evidence for ChoP expression.” (See pg. 20, line 637.)

Table 1. The list of bacteria expressing ChoP modification

ChoP-modified structure	Organisms	Colonization sites	Refs
ChoP-modified glycan ¹			
Teichoic acid	Streptococcus pneumoniae R36A	Respiratory tract	[5]
	Streptococcus oralis Uo5	Respiratory tract	[40]
	Streptococcus mitis NCTC10712	Respiratory tract	[76]
Capsular polysaccharide	Streptococcus pneumoniae type 15	Respiratory tract	[77]
	Streptococcus pneumoniae type 32F	Respiratory tract	[78]
	Erysipelothrix rhusiopathiae	Skin	[44]
Lipopolysaccharides	Haemophilus influenzae Rd	Respiratory tract	[79]
	Haemophilus haemolyticus	Respiratory tract	[36]
	Commensal Neisseria	Respiratory tract	[3]
	Avibacterium paragallinarum	Respiratory tract	[43]
	Pasteurella multocida AP161	Nasopharynx or gastrointestinal tract	[80]
	Histophilus somni 738	Respiratory tract	[7]
	Proteus mirabilis O18	Respiratory, intestinal and urinary tract	[48]
	Morganella morganii O1	Intestinal tract	[49]
Fusobacterium nucleatum strain 25586	Respiratory or intestinal tract	[81]	
Phosphoglycolipid	Mycoplasmas fermentans	Respiratory or urinary tract	[82]
ChoP-modified protein ¹			
Pilus	Neisseria meningitidis	Respiratory tract	[13, 20]
	Neisseria gonorrhoeae	Ocular, nasopharyngeal or anal mucosa	[53, 83]
Fimbrial protein Flp 1	Aggregatibacter actinomycetemcomitans	Respiratory tract	[9]

Porin D	Acinetobacter baumannii	Skin or respiratory tract	[10]
ChoP mimic ¹			
Elongation factor Tu	Pseudomonas aeruginosa	Respiratory tract	[27]
Unknown ChoP-modified structure ²			
Unknown ChoP-modified structure	Bacillus spp.	Gastrointestinal tract	[84]
	Gemella haemolysans	Respiratory tract	[84]
	Micrococcus spp.	Skin	[84]
	Actinomyces viscosus	Oropharynx	[85]
	A. gerencseriae	Oropharynx	
	Lactococcus spp.	Respiratory tract	[84]
	Corynebacterium jeikeium	Skin	[84]
Streptococcus pyogenes	Pharynx, anus, or genital mucosa	[86]	

Footnotes: ¹ Structural evidence for ChoP modification is described in the cited reference, ² studies where mAb recognition of ChoP by TEPEC15 is the only evidence for ChoP expression.

5) It might be useful to cite the review on bacterial ChoP by SE Clark (IAI 2013) which discusses some of the biological advantages of cell surface ChoP expression.

Response

Lines 70-72: As per the reviewer's suggestion, we have now included the citation of SE Clark (IAI 2013) in this manuscript. The sentence now reads as follows:

“The benefits of ChoP modification in bacterial pathogenesis and its impact on the modulation of host immunity have been thoroughly reviewed by Clark and Weiser [15].”
(See pg. 3, lines 70-72.)

6) It might help the readers to explain the structural relationship between PE and ChoP.

Response

Lines 262-265: The structural relationship between PE and ChoP has been elucidated in line 262-265. The revised sentence now reads as follows: “ChoP is PE with three methyl groups. In parasites such as *Plasmodium falciparum* [63] and *Caenorhabditis elegans* [64], PE methyltransferase transfers the three methyl groups to the amine of PE, resulting in the production of ChoP.” (See pg. 8, lines 262-265.)

7) Are there examples of species with both Lic1 and PptA homologs/pathways?

Response

Lines 398-399: Our study revealed that there are no species simultaneously possessing both PptA and Lic-1 homologues in the species containing ChoP modification. We have now added this statement to lines 393-401. The context now reads as follows: “In this study, our analysis of the well-characterised ChoP biosynthetic pathways revealed that the Lic-1 pathway seems to be a common pathway for ChoP-linked glycoconjugates, such as LOS/LPS, WTA, and LTA. In contrast, bacteria that produce ChoP-bearing proteins; such as, *N. meningitidis*, *N. gonorrhoeae*, *A. baumannii*, and *A. actinomycetemcomitans*; appeared to lack the *lic-1* operon, but do contain PptA transferase homologs. There is currently no evidence to suggest the coexistence of the Lic-1 pathway and PptA enzymes within bacterial species. The nature of the pathways that provide the ChoP donor molecule for PptA-dependent transfer of ChoP to ChoP-modified proteins remains to be identified and are the subject of our current on-going studies.”. (See pg. 12, lines 393-401).

8) The potential sources of ChoP (i.e. choline uptake v. biosynthesis) could be explained much more clearly. This should include environmental free choline and the transfer of choline from host intermediates involved in phosphatidylcholine biosynthesis (see Fan X, Mol Micro 2001, 2003) which is not spelled out.

Response

Lines 77-81: As per the suggestion, an explanation about the environmental source for ChoP synthesis was added in lines 77-81. The revised sentence now reads:

“Apart from environmental free choline, choline-containing molecules derived from the host cell lipid metabolism serve as the potential sources for the Lic-1 pathway [17-19]. In the absence of free choline, *H. influenzae* utilizes glycerophosphodiester phosphodiesterase (GlpQ) to acquire choline from the respiratory tract epithelial cells [18, 19].” (See pg. 3, lines 77-81)

9) In Table 1, consider adding the primary site within the host where each species is typically found.

Response

Line 637: As per the suggestion, the primary colonization sites of each specie in the host were added in the Table 1. The revised Table 1 now shows primary colonization sites of each bacterial specie. (See pg. 20, 637.)

ChoP-modified structure	Organisms	Colonization sites	Refs
ChoP-modified glycan ¹			
Teichoic acid	Streptococcus pneumoniae R36A	Respiratory tract	[5]
	Streptococcus oralis Uo5	Respiratory tract	[40]
	Streptococcus mitis NCTC10712	Respiratory tract	[76]
Capsular polysaccharide	Streptococcus pneumoniae type 15	Respiratory tract	[77]
	Streptococcus pneumoniae type 32F	Respiratory tract	[78]
	Erysipelothrix rhusiopathiae	Skin	[44]
Lipopolysaccharides	Haemophilus influenzae Rd	Respiratory tract	[79]
	Haemophilus haemolyticus	Respiratory tract	[36]
	Commensal Neisseria	Respiratory tract	[3]
	Avibacterium paragallinarum	Respiratory tract	[43]
	Pasteurella multocida AP161	Nasopharynx or gastrointestinal tract	[80]
	Histophilus somni 738	Respiratory tract	[7]
	Proteus mirabilis O18	Respiratory, intestinal and urinary tract	[48]
	Morganella morganii O1	Intestinal tract	[49]
Phosphoglycolipid	Fusobacterium nucleatum strain 25586	Respiratory or intestinal tract	[81]
	Mycoplasmas fermentans	Respiratory or urinary tract	[82]
ChoP-modified protein ¹			
Pilus	Neisseria meningitidis	Respiratory tract	[13, 20]
	Neisseria gonorrhoeae	Ocular, nasopharyngeal or anal mucosa	[53, 83]
Fimbrial protein Flp 1	Aggregatibacter actinomycetemcomitans	Respiratory tract	[9]
Porin D	Acinetobacter baumannii	Skin or respiratory tract	[10]
ChoP mimic ¹			
Elongation factor Tu	Pseudomonas aeruginosa	Respiratory tract	[27]
Unknown ChoP-modified structure ²			
Unknown ChoP-	Bacillus spp.	Gastrointestinal tract	[84]

modified structure	Gemella haemolysans	Respiratory tract	[84]
	Micrococcus spp.	Skin	[84]
	Actinomyces viscosus	Oropharynx	[85]
	A. gerencseriae	Oropharynx	
	Lactococcus spp.	Respiratory tract	[84]
	Corynebacterium jeikeium	Skin	[84]
	Streptococcus pyogenes	Pharynx, anus, or genital mucosa	[86]

Footnotes: ¹ Structural evidence for ChoP modification is described in the cited reference, ² studies where mAb recognition of ChoP by TEPEC15 is the only evidence for ChoP expression.

10) A brief discussion of the comparison to phosphocholine biosynthesis to generate PC in eukaryotes would be a useful addition.

Response

The manuscript primarily focuses on prokaryotic bacterial phosphorylcholine. Phosphocholine biosynthesis to generate PC in eukaryotes is beyond the scope of this study.

June 8, 2023

Prof. Michael P. Jennings
Griffith University
Gold Coast Campus, QLD 4222
Australia

Re: Spectrum01583-23R1 (Analysis of bacterial phosphorylcholine related genes reveals an association between the type-specific biosynthesis pathways and the biomolecules targeted for phosphorylcholine modification)

Dear Prof. Michael P. Jennings:

Your manuscript has been accepted, and I am forwarding it to the ASM Journals Department for publication. You will be notified when your proofs are ready to be viewed.

Sincerely,

Xiaoyu Tang
Editor, Microbiology Spectrum
